# Effect of the Ozone Application in the Nutrient Solution and the Yield and Oxidative Stress of Hydroponic Baby Red Chard

Alejandra Machuca Vargas [1], Ana Cecilia Silveira Gómez [2], Cristian Hernández-Adasme [1] and Víctor Hugo Escalona Contreras [1,3,*]

[1] Postharvest Studies Center, Faculty of Agricultural Sciences, University of Chile, Av. Santa Rosa 11315, Santiago 8820808, Chile

[2] Departamento de Producción Vegetal, Facultad de Agronomía, Universidad de la República, Av. Garzón 780, Montevideo 12300, Uruguay

[3] Departamento de Producción Agrícola, Facultad de Ciencias Agronómicas, Universidad de Chile, Av. Santa Rosa 11315, La Pintana, Santiago 8820000, Chile

\* Correspondence: vescalona@uchile.cl

**Abstract:** Novel ozone ($O_3$) sanitizing treatments can be used to decrease the microbial load during cultivation, but they would affect the composition of the nutrient solution. Variations in the nutrient composition decrease crop yields, especially if a strong oxidizing agent such as ozone is used. In this study, $O_3$ was applied throughout the culture every two days at doses of 0.0 (control); 0.5; 1.0; and 2.0 mg·$L^{-1}$ for 3 min on baby red chard (*Beta vulgaris* L. cv. SCR 107) grown in a floating hydroponic system. Macronutrients and micronutrients in the nutrient solution, yield, antioxidant compounds, and oxidative stress enzymes were evaluated in plants. Macronutrients in the nutrient solution were not affected by $O_3$, whereas micronutrients, such as Fe and Mn, decreased by 88.2 and 39.6%, respectively, at the 0.5 mg·$L^{-1}$ dose. The dose of 0.5 mg·$L^{-1}$ produced more fresh matter and leaf area than the control. Antioxidant capacity and total phenols were not significantly affected by $O_3$ treatments; however, higher SOD, CAT, and APX activity after $O_3$ applications were found. It is concluded that ozone applications to the nutrient solution affect the availability of some micronutrients and increase oxidative stress and yield in baby red chard plants.

**Keywords:** floating hydroponic system; baby leaves; sanitizer; food safety; fresh cut

## 1. Introduction

Fresh water scarcity is a worldwide concern and the legislation regarding the use of this resource in highly demanding sectors, such as agriculture, is increasingly strict in terms of safeguarding its availability [1–3]. In this sense, some farmers dedicated to hydroponic crops have implemented systems to reuse the nutrient solution [4,5]. However, the recycling of the solution presents problems such as the loss of nutrients and the proliferation of microorganisms that cause deterioration for plants or human diseases [3,4]. Nevertheless, there is the possibility of applying sanitization methods to guarantee the microbiological quality of the nutrient solution, both for the plants and, above all, for consumers.

Traditionally, there are different types of sanitizations; some are physical methods such as UV-C radiation, slow filtration, thermal treatments (pasteurization), photocatalytic reactions, and other chemicals ones, where ozone ($O_3$) appear as a promoting treatment among others like sodium hypochlorite (NaOCl) [6,7].

Inactivation of microorganisms by $O_3$ is a complex process that affects several components of the membrane and cell wall of plants and microorganisms, as well as enzyme systems and the nucleic acids of the cells [8]. $O_3$ is spontaneously decomposed in the nutrient solution through a series of reaction mechanisms involving the generation of free radicals, termed reactive oxygen species (ROS) [9,10], such as hydroxyl radical ($OH^-$), hydrogen peroxide ($H_2O_2$), and superoxide ($O_2^-$). These ROS are the main oxidizing

agents responsible for the inactivation of microorganisms [11] due to their highly oxidizing nature. Moreover, this oxidative activity can also affect the plants in the hydroponic system, causing irreversible cell damage if ROS are not rapidly removed [12].

However, plants have antioxidant defense mechanisms against oxidative stress, capable of eliminating the ROS produced in the cells. These mechanisms can be classified into enzymatic and non-enzymatic antioxidants. Non-enzymatic antioxidants include polyphenols, ascorbic acid, glutathione, and carotenoids, among others, while enzymatic antioxidants include the enzymes superoxide dismutase (SOD), ascorbate peroxidase (APX) and catalase (CAT) [10]. It should be considered that the oxidative stress caused by ozone at high doses could degrade plant antioxidant compounds due to direct or indirect reactions with ROS [13,14].

On the other hand, the use of $O_3$ in hydroponic systems can be considered as an oxygen supplement to the nutrient solution and an alternative substitute for aeration infrastructure. In fact, some studies support the hypothesis that the application of $O_3$ could saturate nutrient solutions with $O_2$ improving the production of tomatoes [15], Agrostis [4], and bell pepper seedlings [16].

Despite the advantages of $O_3$ compared to other sanitization methods, it is also recognized as a phytotoxic gas [14]. This phytotoxicity has been demonstrated by numerous studies that investigated the response of plants in an $O_3$-enriched troposphere [17], where yield increase and visual quality of plants were affected. However, if $O_3$ is applied in an aqueous medium, the chances of crop deterioration decrease considerably [18].

On the contrary, after reuse of a nutrient solution, the proportions of the nutrients would vary due to the differential ion uptake during the different stages of crop development. As a result, the nutrient solution varies in its ion content, which can lead to deficiencies or toxicities in plants [4]. Previous studies have shown that macronutrients are not affected by ozonation although some micronutrients such as iron (Fe) and manganese (Mn) have precipitated [19,20]. Therefore, although the application of $O_3$ in nutrient solution recycling and sanitization technology has many attractive features, there are also some major disadvantages and a lack of knowledge that prevents farmers from its large-scale application. Therefore, there is a need to re-investigate ozonation treatments as a technology to reuse nutrient solution in hydroponic systems without altering its chemical composition or damaging the vegetable crops. Taking this into account, the objectives of this study were (1) to determine the effect of the $O_3$ applications in a floating root hydroponic system for baby red chard on yield, functional quality, and oxidative stress enzymatic activity, and (2) to evaluate the changes in the nutrient solution exposed to $O_3$ treatments.

## 2. Materials and Methods

### 2.1. Raw Material

Red chard (*Beta vulgaris* L. var. *cicla* cv. SCR 107) characterized by a curly green lamina and red petioles were cultivated in a closed floating-root hydroponic system. The sowing was carried out in trays of 200 alveoli of 18 cm³ polyethylene (Protekta®, Santiago, Chile) on a mixture of an inert substrate composed of rockwool (Agrolan granulado®, Compañía Industrial El Volcán SA, Santiago, Chile) and expanded perlite A6 (Harbolite Chile Ltd.a., Santiago, Chile) hydrated in a volumetric ratio of 2:1. Seedlings were irrigated daily with tap water until the expanded cotyledon stage, which was reached after 10 days from sowing. Subsequently, a modified Hoagland II nutrient solution [21] diluted to 50% was applied. Once the plants reached 2 to 3 true leaves (25 days from sowing), they were transplanted to the closed floating root hydroponic system. This system consisted of 24 polystyrene trays of 20 × 16 × 7 cm containing 1 L of 100% Hoagland II nutrient solution and with 6 plants per tray. Each tray corresponded to one culture unit and three consecutive harvests were made when the leaves reached a length of 8 to 13 cm as a baby stage (after 10, 17, and 26 days from transplant).

The frequency of $O_3$ application and doses used in this experiment was previously determined according to its effect on *Escherichia coli*. For this, a growth curve of this

bacterium was made in the nutritive solution by inoculating $10^6$ cells of *E. coli* (ATCC[®] 35218TM, canine fecal strain of origin. http://www.atcc.org/products/all/35218.aspx, accessed on 4 June 2023). According to this curve, a frequency of $O_3$ application was established every 2 days to decrease the growth of *E. coli* between 2 to 3 logarithmic units following the literature [4,22]. In this way, 4 doses of $O_3$ (0.5; 1.0; 2.0; and 4.0 mg·L$^{-1}$) plus a control (0.0 mg·L$^{-1}$ of $O_3$) were applied for 3 min to achieve a reduction below 2 log CFU·g$^{-1}$, the maximum limit of *E. coli* established by Chilean legislation for fresh cut vegetables [23]. From the five ozone doses evaluated in a pre-test, 4 mg L$^{-1}$ was discarded because it caused notorious phytotoxicity symptoms such as chlorosis, brown spots, and necrosis on the leaves.

The $O_3$ applications to the nutritive solution began to be carried out after 5 days of transplant, using doses of 0.0 (D0); 0.5 (D05); 1.0 (D1); and 2.0 (D2) mg·L$^{-1}$ for 3 min every 2 days throughout the crop cycle. Additionally, all culture units were aerated every 3 h for 3 min during the plant cultivation using aquarium air pumps and time controllers. The pH of the nutrient solution was monitored throughout the experiment with a potentiometer (Hanna Instruments, pH21, Romania) and adjusted to a pH of 5.5–5.8 with a solution of nitric and phosphoric acids. Electrical conductivity was monitored with a conductivity meter (Hanna Instruments, HI 99301, Romania).

An $O_3$ generator (Atlas 30 g, Ecozone, AB, Canada) was used and $O_3$ gas was obtained from dry air and an electric discharge generator. The $O_3$ concentration was determined with a portable $O_3$ analyzer (Chemetrics, model I-2019, Midland, TX, USA).

*2.2. Plant Tissue Evaluations*

Plant tissue evaluations were performed at each harvest (after 10, 17, and 26 days from transplant). The samples of each repetition for the chlorophyll and oxidative stress determinations were kept in a −80 °C freezer until analysis.

Fresh weight. An analytical balance (Radwag, AS 100, Radom, Poland) was used. To obtain the mass, the entire aerial part was extracted by cutting at the height of the neck of the plant, separating it from the root part. The results were expressed in grams (g).

Dry weight percentage. Fresh samples were dried in an oven with forced air ventilation at 70 °C (LabTech, model LDO-50F, Santiago, Chile) until a constant weight was obtained. The results were expressed as a percentage of dry weight (%) by means of the difference between the initial and final weights.

Leaf area. In each harvest, the leaf area was determined with the Sigma Scan pro 5.0 software, and the results were expressed in cm$^2$.

Chlorophyll *a* and *b* contents. A total of 0.4 g of frozen sample (distal sector of the lamina without rib and petiole) were homogenized with 5 mL of acetone (80%), using an Ultraturrax (IKA, T18 basic, Staufen, Germany) for 45 s. For quantification, the Lichtenthaler and Wellburn method was used (1983). The absorbance of the extract was measured at wavelengths of 646 and 663 nm using a UV-Vis spectrophotometer (UV-vis, T70, PG Instruments Limited, Leicestershire, UK). For the quantification, the following expressions were used [24]:

$$Ca = 12.25\ A663 - 2.79\ A646 \qquad (1)$$

$$Cb = 21.5\ A646 - 5.1\ A663 \qquad (2)$$

where:

C*a*: Chlorophyll *a* content (mg·g$^{-1}$ fresh weight, fw)
C*b*: Chlorophyll *b* content (mg·g$^{-1}$ fresh weight, fw)
A663: sample absorbance measured at 663 nm
A646: sample absorbance measured at 646 nm

*2.3. Determinations of the Nutritive Solution*

The following determinations of the physical and chemical parameters of the nutrient solution were carried out.

Chemical analysis of micro and macronutrients. $Cl^-$, $NO_3^-$ and $NH_4^+$ were determined with potentiometry; $Ca^{2+}$, $Mg^{2+}$, $K^+$, $Fe^{2+}$, $Mn^{2+}$, $Zn^{2+}$, and $Cu^{2+}$ were determined with atomic absorption spectrophotometry according to the methodology described by Sadzawka et al. [25]; while $B^{+3}$ and $SO_4^{2-}$ were determined through the colorimetric method; and $HCO_3^-$ was determined with the volumetric method [26]. This analysis was performed in an independent trial with nutrient solution only, and chemical analysis was accomplished before and 20 min after ozone application with the lowest dose of $O_3$ (0.5 mg·$L^{-1}$). These analyses were performed independently to avoid interaction between the fresh tissue, such as the roots, and the ozone.

Dissolved oxygen. It was measured daily in the nutrient solution with a dissolved oxygen meter (Hanna Instruments, HI 9146, Nusfalau, Romania) and expressed in mg·$L^{-1}$.

*2.4. Determinations Associated with Oxidative Stress in the Leaves*

Total phenols content. It was determined with the method of Singleton and Rossi [27] with some modifications. For the preparation of the extract, 5 g of leaves were weighed in a 50 mL Falcon tube, and 20 mL of methanol was added. It was homogenized for 30 s with a homogenizer (IKA, T18 basic, Staufen, Germany) and kept in the darkness for 24 h at 5 °C. Subsequently, the supernatant was filtered through filter paper (grade 292) and centrifuged for 15 min at 3840× $g$ at 4 °C. A total of 0.5 mL of extract was taken, and 8 mL of miMilli-Q water was added. Subsequently, 0.5 mL of 0.25 N Folin Ciocalteu reagent was added and left to react for 3 min. Then 1 mL of 1 N $Na_2CO_3$ was added and allowed to react for 10 min. The absorbance of the samples was measured at a wavelength of 725 nm using a microplate spectrophotometer (Biochrom, UVM 340, Cambridge, UK). A calibration curve was made with gallic acid. Total phenols content was expressed as gallic acid equivalent per g of fresh weight (GAE in mg·$g^{-1}$ fw).

### 2.4.1. Antioxidant Capacity

The antioxidant activity of the extracts was determined using the 2,2-diphenyl-1-picryl hydrazyl hydrate violet radical (DPPH) [28] and the iron reduction antioxidant power (FRAP) methods, both with modifications [29]. The extraction of the samples was carried out following the same methodology used for total phenols. A total of 194 μL of the DPPH solution was added to 21 μL of extract and the absorbance was measured at 517 nm using a microplate spectrophotometer after waiting 30 min at room temperature (Biochrom Asys UVM 340, Biochrom, Cambridge, UK). The antioxidant activity was also evaluated by adding 198 μL of the FRAP reagent to 6 μL of extract. Absorbance was measured at 593 nm after waiting 30 min at room temperature using a microplate spectrophotometer (Biochrom Asys UVM 340, Biochrom, Cambridge, UK). The results in both methods were expressed as equivalent mg of Trolox per g of fresh weight (mg TE·$g^{-1}$ fw).

### 2.4.2. Oxidative Stress Enzymes

Catalase (CAT). CAT activity was measured following the protocol described by Lemoine et al. [30] with modifications. The enzyme activity was determined by the decomposition of $H_2O_2$ at 240 nm. For the enzyme extraction, one gram of leaves was weighed in 50 mL Falcon tubes, then 0.2 g of polyvinylpyrrolidone (Sigma-Aldrich, St. Louis, MO, USA) and 5 mL of extraction buffer composed of 50 mM potassium phosphate buffer at pH 7.8 (Merck, Darmstadt, Germany), 0.1 mM ethylenediaminetetraacetic acid (EDTA) (Merck, Darmstadt, Germany), 5 mM L-cysteine (Sigma-Aldrich), 0.2% Triton X-100 (Merck, Calbiochem, Darmstadt, Germany), and 50 μL of phenyl sulfonyl fluoride (Merck, Darmstadt, Germany) were added. Subsequently, the mixture of reagents and leaves was homogenized with an Ultraturrax (IKA, T18 basic, Staufen, Germany) for 30 s, keeping the tubes in an ice bath to avoid heating of the samples. For the measurement, 50 mL of 50 mM potassium

phosphate buffer at pH 7 with 10.6 mM $H_2O_2$ was prepared. Then, in a quartz plate, 50 μL of the enzyme extract and 1450 μL of reaction buffer were added. Absorbance was measured at 240 nm for 5 min in a spectrophotometer (UV-vis, T70, PG Instruments Limited, Leicestershire, UK). The increase in absorbance of the linear part was taken. To determine the reaction rate of the non-enzymatic oxidation of $H_2O_2$, a measurement was made with only the reaction buffer. The enzymatic activity was expressed by activity units (AU) per mg of protein.

Ascorbate peroxidase (APX). The principle of the measurement of this enzyme is given by the reduction in absorbance due to the oxidation of ascorbic acid [31]. The extract was the same that was used for CAT. For the measurement, 50 mL of reaction buffer composed of 50 mM potassium phosphate buffer at pH 7.0 (Merck, Darmstadt, Germany), 0.1 mM EDTA (Merck, Darmstadt, Germany), 0.5 mM ascorbic acid (Merck, Darmstadt, Germany), and 1.54 mM $H_2O_2$ (Merck, Darmstadt, Germany) were mixed. Then, in a quartz plate, 505 μL of enzyme extract plus 960 μL of reaction buffer was added. Samples were measured at 290 nm every 30 s in a spectrophotometer (UV-vis, T70, PG Instruments Limited, Leicestershire, UK) until the absorbance remained constant. The enzymatic activity was expressed by activity units (AU) per mg of protein.

Super oxide dismutase (SOD). SOD activity was determined by the ability to inhibit the photochemical reduction of nitro blue tetrazolium (NBT) using the method described by Dhindsa et al. [32] with slight modifications. For enzyme extraction, 3.5 g of leaves were weighed in 50 mL Falcon tubes, then 5 mL of Tris-HCL extraction buffer at pH 7.5, 3 mM $MgCl_2$ (Merck, Darmstadt, Germany), and 1 mM EDTA were added (Merck, Darmstadt, Germany). Subsequently, the sample was homogenized for 1 min and centrifuged at $3840\times g$ for 20 min at 4 °C. For the measurement, 100 mL of reaction buffer was prepared with a 50 mM potassium phosphate at pH 7.8 (Merck, Darmstadt, Germany), 13 mM methionine (Merck, Darmstadt, Germany), 75 mM NBT (Merck, Chile), 2 μM riboflavin (Merck, Darmstadt, Germany), and 0.1 M EDTA (Merck, Darmstadt, Germany). A total of 6 μL of enzyme extract and 351 μL of reaction buffer were added to each well per plate. To achieve photoreduction of NBT, plates were exposed to 15 W for 15 min. Absorbance was measured at 560 nm in a spectrophotometer (Biochrom Asys UVM 340, Biochrom, Cambridge, UK) and enzyme activity was expressed as activity units (AU) per mg of protein. Protein quantification was performed using the Bradford method [33]. A total of 5 μL of extract and 250 μL of Bradford reagent (Merck, Darmstadt, Germany) were mixed. The mixture was allowed to react for 30 min and measured at 595 nm for absorbance. Protein concentration was determined using a standard curve for each enzyme with bovine serum albumin (Merck, Darmstadt, Germany).

### 2.5. Experimental Design and Statistical Analysis

A completely randomized experimental design with a $4 \times 3$ factorial arrangement was used, where the first factor consisted of four $O_3$ doses and the second, in three consecutive harvests, with 3 repetitions per treatment. The experimental unit corresponded to a cultivation unit with 6 plants each. For the experiment on the effect of $O_3$ in the nutrient solution, a completely randomized design with 2 treatments (with or without $O_3$) was used, and 3 replicates per treatment were established. In this case, an experimental unit corresponded to a container with 1 L of nutrient solution. The results were analyzed with analysis of variance (ANDEVA). For the statistical analysis, Infostat (version 2015, FCA, Universidad Nacional de Cordoba, Córdoba, Argentina) software was used. Tukey's multiple comparison test was applied to analyze the differences between the parameters measured at different levels of significance.

## 3. Results and Discussion

### 3.1. Evaluation of Plant Growth

Fresh weight. The fresh weight of the chard fluctuated between 10.25 and 15.14 g throughout the experiment (Table 1). According to the analysis of the data, it was observed

that the dose factor presented significant differences ($p \leq 0.0001$). In relation to these results, the D05 treatment differed significantly from the rest of the treatments, including the control, obtaining the highest average value in fresh weight of 14.35 g (Table 2). No differences in fresh weights were found between consecutive harvests, probably because leaf length was used as a harvest index. Because of this, the harvest periods were not equal. In addition, the experiments and harvests were conducted consecutively between autumn and winter, so lower maximum and minimum temperatures and shortened day length could have influenced plant performance.

**Table 1.** Fresh weight (g) of hydroponic baby red chard cv. SCR 107 exposed to four $O_3$ doses and three consecutive harvests.

| $O_3$ Dose (mg·L$^{-1}$) | Fresh Weight (g) | | |
|:---:|:---:|:---:|:---:|
| | Harvest 1 (day 10) | Harvest 2 (day 17) | Harvest 3 (day 26) |
| D0 | 11.56 ± 0.38 [1] | 10.86 ± 0.84 | 11.56 ± 0.38 |
| D05 | 14.79 ± 0.49 | 15.14 ± 0.25 | 13.12 ± 0.39 |
| D1 | 11.13 ± 0.12 | 12.06 ± 0.76 | 11.13 ± 0.12 |
| D2 | 10.25 ± 0.28 | 11.73 ± 0.41 | 11.25 ± 0.13 |
| | Significance level | | |
| $O_3$ dose (D) | **** (1.49) [2,3] | | |
| Harvest time (H) | NS [3] | | |
| D × H | NS | | |

[1] The values correspond to means ($n$ = 3) ± standard error. [2] MSD (Minimal significant difference). [3] NS, ****. Not significant or significant for $p \leq 0.05$, 0.01, 0.001 or 0.0001, respectively.

**Table 2.** Fresh weight (g) of hydroponic baby red chard cv. SCR 107 exposed to four $O_3$ doses.

| $O_3$ Dose (mg·L$^{-1}$) | Fresh Weight (g) |
|:---:|:---:|
| D0 | 11.63 b [1] |
| D05 | 14.35 a |
| D1 | 11.32 b |
| D2 | 11.44 b |

[1] Different letters indicate significant differences for the $O_3$ dose factor according to Tukey's test ($p$-value $\leq 0.0001$). Values represent the mean of the three consecutive harvests.

Dry weight percentage. No significant differences were found for harvest time nor for the interaction between factors (Table 3). The $O_3$ dose factor showed significant differences ($p \leq 0.05$) in the dry weight percentage parameter. The means for each dose were between a range of 11.56 and 13.82%, significantly differing between the D05 treatment and the control (D0) (Table 4).

Leaf area. Values ranged between 8.98 and 15.21 cm$^2$ (Table 5) in the three consecutive harvests carried out. Significant differences were found in the ozone dose factor. D05 treatment induced a significantly higher leaf area production compared to the rest of the treatments (Table 6). D2 also presented significant differences compared to the control.

Biomass, measured as fresh matter and leaf area, is the most relevant parameter of the impact caused by stress depending on the general plant state. The irrigation ozone application can provoke oxidative stress depending on the plant's physiological stage which positively or negatively affects growing conditions and the crops [14]. $O_3$ induces various processes that would explain a decrease in photosynthesis, such as stomatal closure and, consequently, a decrease in $CO_2$ fixation and rubisco enzyme activity and concentration, among others [10,34]. Although in this experiment some of the $O_3$ artificially applied to the nutrient solution could escape as a gas to the atmosphere by bubbling, direct contact with the leaves resulted in lower yield at high $O_3$ doses. According to Graham et al. [4],

a higher $O_3$ concentration of 3 mg $L^{-1}$ applied to the nutrient solution achieved a higher residual gas in the tomato crop and, consequently, lower leaf area.

**Table 3.** Dry weight percentage of hydroponic baby red chard cv. SCR 107 exposed to four $O_3$ doses and three consecutive harvests.

| $O_3$ Dose (mg·$L^{-1}$) | Dry Weight (%) | | |
|---|---|---|---|
| | Harvest 1 (day 10) | Harvest 2 (day 17) | Harvest 3 (day 26) |
| D0 | 15.23 ± 0.71 [1] | 12.87 ± 0.39 | 13.37 ± 0.27 |
| D05 | 12.87 ± 1.07 | 10.17 ± 0.23 | 11.63 ± 0.22 |
| D1 | 11.88 ± 0.63 | 11.85 ± 0.30 | 12.45 ± 0.39 |
| D2 | 14.52 ± 1.65 | 13.31 ± 0.36 | 13.10 ± 0.23 |
| Significance level | | | |
| $O_3$ dose (D) | * (1.49) [2] | | |
| Harvest time (H) | NS [3] | | |
| D × H | NS | | |

[1] The values correspond to means ($n$ = 3) ± standard error. [2] MSD (Minimal significant difference). [3] NS, *: Not significant or significant for $p \leq 0.05$.

**Table 4.** Dry weight (%) of hydroponic baby red chard cv. SCR 107 exposed to four $O_3$ doses.

| $O_3$ Dose (mg·$L^{-1}$) | Dry Weight (%) |
|---|---|
| D0 | 13.82 a [1] |
| D05 | 11.56 b |
| D1 | 12.06 ab |
| D2 | 13.65 ab |

[1] Different letters indicate significant differences according to Tukey's test ($p$-value ≤ 0.05). Values represent the means ($n$ = 9) of the three consecutive harvests.

**Table 5.** Leaf area ($cm^2$) of hydroponic baby red chard cv. SCR 107 exposed to four $O_3$ doses and three consecutive harvests.

| $O_3$ Dose (mg·$L^{-1}$) | Leaf Area ($cm^2$) | | |
|---|---|---|---|
| | Harvest 1 (day 10) | Harvest 2 (day 17) | Harvest 3 (day 26) |
| D0 | 11.44 ± 0.20 [1] | 11.02 ± 0.37 | 10.83 ± 0.62 |
| D05 | 15.00 ± 0.40 | 15.21 ± 0.26 | 14.91 ± 0.35 |
| D1 | 12.78 ± 0.78 | 12.21 ± 0.25 | 12.10 ± 0.36 |
| D2 | 8.98 ± 0.38 | 11.97 ± 0.61 | 10.71 ± 0.13 |
| Significance level | | | |
| $O_3$ dose (D) | **** (1.68) [2] | | |
| Harvest time (H) | NS [3] | | |
| D x H | NS | | |

[1] The values correspond to means ($n$ = 3) ± standard error. [2] MSD (Minimum Significant Difference). [3] NS, ****. Not significant or significant for $p \leq 0.0001$, respectively.

**Table 6.** Leaf area (cm$^2$) of hydroponic baby red chard cv. SCR 107 exposed to four O$_3$ doses.

| O$_3$ Dose (mg·L$^{-1}$) | Leaf Area (cm$^2$) |
|---|---|
| D0 | 10.55 a [1] |
| D05 | 15.04 c |
| D1 | 11.09 ab |
| D2 | 12.37 b |

[1] Different letters indicate significant differences according to Tukey's test (*p*-value $\leq$ 0.001). Values represent the means (*n* = 9) of the three consecutive harvests.

In this study, an increase in yield expressed as fresh weight and leaf area was observed with 0.5 mg·L$^{-1}$ of O$_3$ compared to the rest of the treatments, including the control. Similar results were found under O$_3$ applications of 7 g·h$^{-1}$ to the lettuce cv. 'Subyana F1' seeds, which increased yield by 8.66% (evaluated as g per lettuce) [35]. In another study, Malaiyandi and Natarajan [36] observed increases in dry weight, shoot length, and leaf area of cowpeas after exposure to 0.28 mg·L$^{-1}$ of O$_3$. This may suggest that O$_3$ can stimulate plant growth under certain conditions and plant response could be species or even cultivar related. This could be very interesting for the future use of certain concentrations of O$_3$ as it could be recommended to initiate growth in certain species or physiological phases [35].

Chlorophyll *a*. The Chl*a* range in the leaves fluctuated between 0.49 and 1.02 mg g$^{-1}$ fw, its concentration was affected by the interaction between the O$_3$ dose and harvest time factors (*p* $\leq$ 0.001) (Table 7). In harvest 1, a decrease in Chl*a* was observed, with significant differences between D1 and D2 treatments compared to the control (D0). In harvest 2, only D2 treatment varied from D05, while in harvest 3, both D1 and D2 differed from the control. Regarding the dose factor, in D2 treatment after harvests 2 and 3 significant differences were found compared to harvest 1.

**Table 7.** Effect of the interaction between the O$_3$ doses and consecutives harvests on the concentration of chlorophylls *a* and *b* (mg g$^{-1}$ fw) of hydroponic baby red chard cv. SCR 107.

| | O$_3$ Dose (mg·L$^{-1}$) | Harvest 1 (Day 10) | Harvest 2 (Day 17) | Harvest 3 (Day 26) |
|---|---|---|---|---|
| | D0 | 1.02 Aa [1,2] | 0.80 Ba | 0.81 Aa |
| | D05 | 0.79 Aa | 0.70 Ba | 0.76 Aa |
| Chl*a* | D1 | 0.63 BCb | 0.86 Aba | 0.69 Bab |
| | D2 | 0.49 Cc | 0.95 Aa | 0.72 Bb |
| | D0 | 1.34 Aba | 1.33 Ba | 1.34 Aa |
| | D05 | 1.33 Bb | 1.35 Aba | 1.33 Aa |
| Chl*b* | D1 | 1.32 Bb | 1.36 Aa | 1.34 Aa |
| | D2 | 1.36 Aa | 1.35 Aa | 1.34 Aa |

[1] Values correspond to means (*n* = 3). [2] Means followed by different letters, uppercase for columns and lowercase for rows, are significant differences according to Tukey's test (*p* $\leq$ 0.0001).

Chlorophyll *b*. The range of Chl*b* was 1.32 and 1.36 mg g$^{-1}$ fw, observing an interaction between O$_3$ dose and harvest factors (*p* $\leq$ 0.001) (Table 7) following a trend like Chl*a*. In harvest 1, the D2 and D0 doses obtained higher values and contrasted with the D05 and D1 doses. In harvest 2, D1 and D2 treatments varied from the control, while in harvest 3 no significant differences were observed. In the analysis by dose, with D05 treatment, harvest 1 was significantly distinguished from harvests 2 and 3. D1 treatment at harvest 1 was significantly different from harvests 2 and 3, while with D2 treatment no significant differences were observed among harvests.

In studies on the effect of increased atmospheric O$_3$ on these chlorophylls, a decrease in their concentrations has been observed in bean, soybean, potato, and lettuce plants, due to the high oxidative power of O$_3$ [18,37–39]. In fact, studies in pea (*Pisum sativum* L.) cvs.

Little Marvel Perfection and Victory subjected to $O_3$ exposures between 70 to 90 nL·L$^{-1}$ for 8 h·day$^{-1}$ showed 19, 16, and 30% decreases in Chl*a*, respectively [40].

As is known, chlorophyll plays an important role in capturing light for photosystem I and II that provide energy-rich molecules (ATP and NADPH) in the Calvin cycle during photosynthetic electron transport in the thylakoid membrane [41]. Consequently, if there is a detriment to chlorophyll, photosynthesis and plant yield can be affected [41]. In the present study, an interaction was found between the $O_3$ doses and the harvest times in both types of chlorophylls, which would indicate that both were jointly affected by these two factors. This decrease in the content of chlorophylls could explain the reduced fresh weight, percentage of dry weight, and the leaf area found in those plants exposed to the highest $O_3$ doses.

Another important aspect to consider is the visible leaf damage, which was observed after applications of 2 mg·L$^{-1}$ of $O_3$. Chlorosis, brown spots, and even necrosis observed in certain sectors of the leaves could be explained by the decrease in Fe content found at the end of the experiment in the nutrient solution (Figure 1). As mentioned, Fe plays a fundamental role in the composition of plant chlorophylls [20].

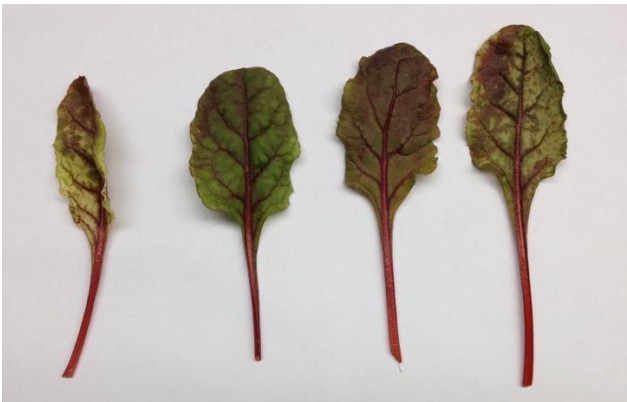

**Figure 1.** Visual symptoms on hydroponic baby red chard leaves at harvest (on day 26) after $O_3$ applications of 2 mg·L$^{-1}$ every 2 days.

Similarly, tomato plants intermittently irrigated with ozonated water showed significantly lower Fe content in leaf tissue compared to control and continuous irrigation with ozonated water [17]. In a study on rice, brown spots were observed after 24 h of an $O_3$ application of 150 g·L$^{-1}$ for 6 h, produced by the generation of $H_2O_2$ (as a product of the $O_3$ application) that diffuses through the apoplast and induces responses such as lipid peroxidation [12]. This ozone dose could be considered high compared to the other studies mentioned before, which with lower concentrations observed visible leaf damage [40]. This point is of special interest for commercial organs such as leaves, like lettuce, spinach, or chard, where the presence of visible symptoms causes the immediate depreciation of the product. In this sense, Gottardini et al. [42] observed a decrease in Chl*a* even before observing visible damage in lantana plants (*Viburnum lantana* L.) exposed to 0.04 mg·L$^{-1}$ of $O_3$ due to oxidative stress damage, which could provide a diagnostic indicator for the early assessment of possible $O_3$ effects on plants.

### 3.2. Chemical Analysis in the Nutrient Solution

Based on the results obtained for leaf area and fresh matter, an independent test was conducted to determine the effect of the $O_3$ application on the macronutrient and micronutrient contents of the nutrient solution. In fact, a low ozone dose (D05) applied only once was evaluated because the highest dose caused visible damage to plant tissue, ruling out its evaluation in the nutrient solution.

Micronutrients. The application of 0.5 mg L$^{-1}$ of $O_3$ caused a significant reduction of $Fe^{2+}$ and $Mn^{2+}$ of 88.2 and 39.6% ($p \leq 0.05$), respectively, at 20 min after application

compared to the control (Figure 2). These two microelements were the most reduced by $O_3$ application. On the other hand, the microelements boron ($B^{+3}$) and copper ($Cu^{+2}$) were reduced in the range of 2.9 and 10%, while Zinc ($Zn^{+2}$) showed a slight increase of 2.1%. According to these results, oxidation of $Fe^{2+}$ from solution due to ozonation is a significant concern, particularly in hydroponic systems. Particularly since iron (Fe) is water soluble and readily available to plants as ferrous ion ($Fe^{2+}$) [43]. However, agreeing to some authors, these losses could be largely controlled with the use of iron chelates, such as ethylenediaminetetraacetic acid (EDTA), diethylenetriaminepentaacetic acid (DTPA) or ethylenediamine-N,N′-bis (2-hydroxyphenylacetic acid) (EDDHA) [20,44]. These chelating compounds form a soluble complex with cations held by ionic forces so that they are readily available to plants [45]. In this study, the form of Fe used in the nutrient solution was a mixture of Fe chelates (EDTA and EDDHA). However, despite the use of these chelates, the oxidation of this microelement was significant. On the other hand, the rate of Fe oxidation by $O_3$ is reduced when the pH of the nutrient solution is lower than 7.5 [43]. Although the pH of the nutrient solution during this study fluctuated between 5.5 and 5.8, the loss of Fe was almost 40% due to ozonation.

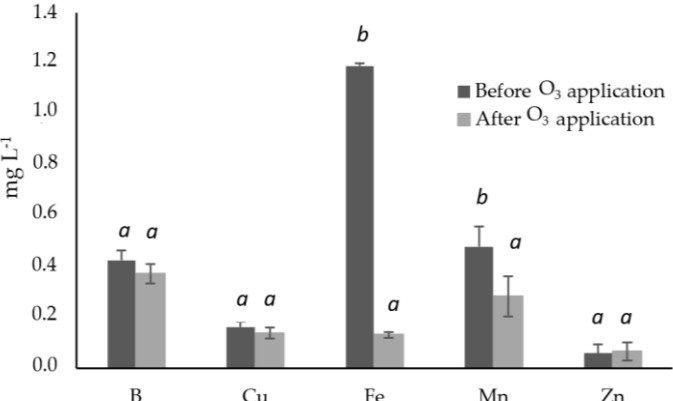

**Figure 2.** Micronutrient content (mg $L^{-1}$) in the nutrient solution before and after applying 0.5 mg·$L^{-1}$ of $O_3$. Different letters indicate significant differences in each micronutrient according to Tukey's test ($p$-value $\leq$ 0.05). Values represent the mean ($n$ = 3) $\pm$ standard error.

Fe acts as a cofactor in the photosynthetic electron transport chain and is essential for chlorophyll biosynthesis. In fact, chloroplasts contain up to 90% of the Fe found in leaf cells, with approximately half in the stroma and the rest in the thylakoid membranes [46]. Therefore, a deficiency of Fe can result in strong chlorosis that usually begins to develop in the youngest leaves [47]. In addition, Fe is also a component of many enzymes associated with energy transfer, nitrogen reduction and fixation, and lignin formation [48,49].

Due to the multiple and important functions of Fe in plants mentioned above, some authors point out that a deficiency of Fe could result in a decrease in biomass production [50–52]. However, other authors indicated that a moderate deficiency could result in an increase in biomass. For example, in a study conducted on spinach exposed to moderate deficiencies of this micronutrient (1–10 μM), an increase in fresh matter as well as in nutritional quality in terms of sugars, proteins, and nitrates was obtained as a result [53]. The reasons that would explain this increase in yield despite the detriment of Fe lie in the activation of the LeKC1 and LePT1 genes responsible for encoding the $K^+$ channels and the phosphate transporter, respectively [54]. Similarly, the LeNRT2.1 and LeNRT1.2 genes encode transport media for nitrates facilitating the absorption uptake of N, P, and K by the roots and, consequently, an increase in biomass [55].

As for Mn, it is the second most needed microelement in plants after Fe, and is involved in biological processes such as photosynthesis, respiration, and assimilation of nitrogen [56]. If the ozonation is excessive or prolonged, the Mn can be oxidized [43]. The main symptom

of Mn deficiency is interveinal chlorosis associated with the development of necrotic spots observed on old and young leaves depending on the species and growth rate [56].

The normal range in the nutrient solution for Mn is between 0.25 to 0.5 mg·L$^{-1}$ and for Fe from 0.5 to 1.5 mg·L$^{-1}$ [57]. Considering these values, Mn maintained an acceptable range after ozonation (0.28 mg·L$^{-1}$), while Fe decreased due to precipitation to deficiency ranges (0.13 mg·L$^{-1}$). Thus, if the loss of this element due to ozonation is high, a replacement of this element after the O$_3$ application should be considered if crop damage warrants it. The values obtained for Fe in this study would indicate that deficiency of this micronutrient can affect the yield of chard when doses of 0.5 mg·L$^{-1}$ of O$_3$ are used; however, no yield reduction was observed with respect to the control in any of the crops.

Macronutrients. Ca$^{2+}$, Mg$^{2+}$, K$^+$, Na, Cl$^-$, SO$_4$, HCO$_3$, NH$_4{}^+$, NO$_3{}^-$, and H$_2$PO$_4{}^-$ were not significantly affected by the application of 0.5 mg·L$^{-1}$ of O$_3$, decreasing between 4.4 and 6.36% (Figure 3), which agrees with the existing literature. Thus, a study published by Vernon [58] reported no significant losses in these macroelements due to the application of O$_3$ at concentrations of 0.0; 0.5; 1.0; and 1.5 mg·L$^{-1}$.

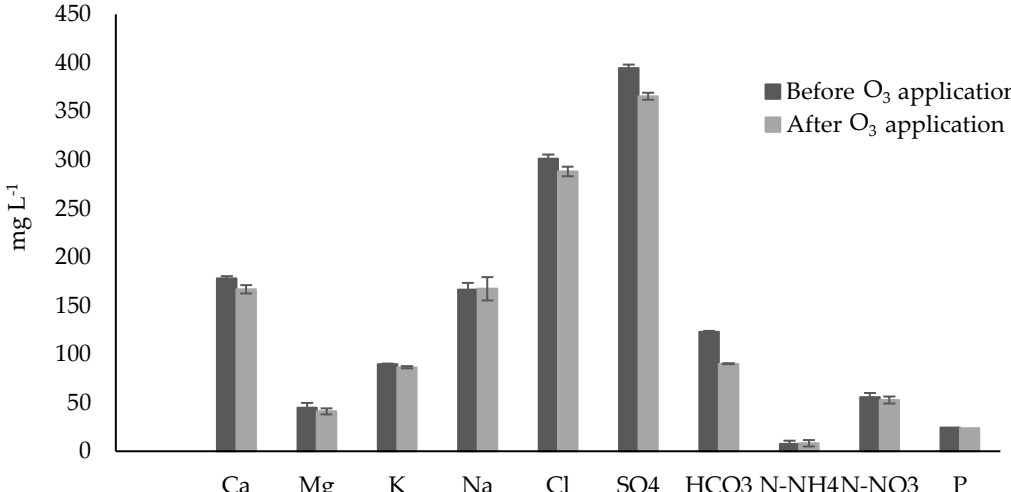

**Figure 3.** Macronutrient content (mg L$^{-1}$) in the nutrient solution before and after applying 0.5 mg·L$^{-1}$ of O$_3$. Values represent the mean ($n$ = 3) ± standard error.

Dissolved oxygen. The dissolved oxygen range fluctuated between 5.68 and 10.86 mg·L$^{-1}$. The highest values of this parameter were observed with the D2 dose. As expected, with the O$_3$ treatments, oxygen was significantly higher than in the control treatment. A normal range of oxygenation in a hydroponic culture is considered to be between 6 and 8 mg·L$^{-1}$ at 20 °C [14]. The highest dose of O$_3$ applied (D2) achieved the highest O$_2$ concentration in the nutrient solution during cultivation (10 mg·L$^{-1}$). The rest of the treatments, including the control, obtained concentrations between 6 and 9 mg·L$^{-1}$. In a hydroponic tomato study conducted by Graham et al. [4], as a consequence of the application of 3 mg·L$^{-1}$ of O$_3$ to the nutrient solution, a significant increase in dissolved oxygen up to 18.4 mg·L$^{-1}$ was observed. Also, with this dose, an increase in leaf area was achieved. In this sense, the authors remarked that it seems that O$_3$ has a stimulating effect on plant growth, especially in the initial growth phase, since significant increases in leaf area were found during the vegetative development stage, although these were not observed in tomato fruits. These results are in agreement with those obtained in our study, since O$_3$ treatments achieved a significant increase in leaf area, as previously reported.

*3.3. Oxidative Stress in the Leaves*

3.3.1. Total Phenol Content

After harvest 1, the values were 0.88 to 0.94 mg GAE·g$^{-1}$; while at harvest 2, the range was from 0.87 to 0.92 mg GAE·g$^{-1}$; and at the end of harvest 3, values of 0.86 to

0.90 mg GAE$\cdot$g$^{-1}$ were recorded (Table 8). Neither O$_3$ application nor harvest day affected total phenol content ($p = 0.4725$). According to the background literature reviewed, there are no studies evaluating the effect of O$_3$ applied to the nutrient solution on phenolic compounds in hydroponic crops. There are reports of increases in phenolic compounds in response to increased atmospheric O$_3$, as in the case of red clover (*Trifolium pratense* L.), whose total phenolic compounds increased between 5 and 12% with concentrations of 0.038 and 0.056 mg$\cdot$L$^{-1}$ of O$_3$ (range 1.26 times above ambient O$_3$) [59]. In wheat (*Triticum aestivum* L.) exposed to 0.384 mg$\cdot$L$^{-1}$ of O$_3$, an increase in phenol content was also observed; however, this increase depended on the phenological stage of the crop, being higher at the grain-filling stage than during vegetative growth [60]. In another study of red lettuce grown under artificial light, the total phenolics, flavonoids, and anthocyanins showed a significant increase after 0.25 h of exposition on the fourth leaf [61]. The above could partly explain our results since the O$_3$ applications and the analyzes were performed at an early stage of the plant, after 5 days post-transplantation.

**Table 8.** Total phenol content (mg GAE$\cdot$g$^{-1}$ fw) of hydroponic baby red chard cv. SCR 107 exposed to four O$_3$ doses at three consecutive harvests.

| O$_3$ Dose (mg$\cdot$L$^{-1}$) | Total Phenol Content (mg GAE$\cdot$g$^{-1}$ fw) | | |
|---|---|---|---|
| | Harvest 1 (day 10) | Harvest 2 (day 17) | Harvest 3 (day 26) |
| D0 | 0.91 $\pm$ 0.07 [1] | 0.92 $\pm$ 0.03 | 0.88 $\pm$ 0.04 |
| D05 | 0.88 $\pm$ 0.05 | 0.87 $\pm$ 0.05 | 0.90 $\pm$ 0.03 |
| D1 | 0.91 $\pm$ 0.06 | 0.89 $\pm$ 0.02 | 0.86 $\pm$ 0.04 |
| D2 | 0.94 $\pm$ 0.05 | 0.90 $\pm$ 0.07 | 0.87 $\pm$ 0.05 |
| | Significance level | | |
| O$_3$ dose (D) | NS [2] | | |
| Harvest time (H) | NS | | |
| D $\times$ H | NS | | |

[1] The values correspond to means ($n = 3$) $\pm$ standard error. [2] NS: Not significant.

### 3.3.2. Antioxidant Capacity

The values evaluated with the FRAP method fluctuated between 0.95 and 1.20 mg TE$\cdot$g$^{-1}$ (harvest 1), 1.02 and 1.05 mg TE$\cdot$g$^{-1}$ (harvest 2), and 0.98 and 1.00 mg TE$\cdot$g$^{-1}$ (harvest 3) (Figure 4A) without significant differences among doses or days of harvest ($p = 0.2579$). While the mean, as measured via the DPPH method, in baby red chard harvested at different times reached between 0.81 and 0.89 mg TE$\cdot$g$^{-1}$ (harvest 1), 0.84 and 0.91 mg TE$\cdot$g$^{-1}$ (harvest 2), and 0.80 and 0.89 mg TE$\cdot$g$^{-1}$ (harvest 3), with no significant differences among treatments either ($p = 0.1288$) (Figure 4B). As in the present study, in beans (*Phaseolus vulgaris* L.) treated with 0.123 mg$\cdot$L$^{-1}$ of O$_3$ for one week, no significant changes were observed in the antioxidant capacity of the leaves measured with the FRAP method, which determines the non-enzymatic antioxidant capacity, expressed as the total content of the thiol group [62]. In contrast, in a study on clover (*Trifolium repens* L.) exposed to O$_3$ (100 Nl$\cdot$L$^{-1}$), a 7- to 9-fold increase in antioxidant capacity (by FRAP) was observed relative to untreated plants [63]. In the present study, no significant variations in the antioxidant capacity (by FRAP and DPPH methods) were found, which could indicate that antioxidants of non-enzymatic origin are not altered with the doses and frequency evaluated. The response to ozone in red lettuce varied as a function of gas concentration and the leaf age. The total phenol and flavonoid content and antioxidant capacity of the third leaves exposed to 100 ppb of ozone were like the control but in the fourth leaf, the antioxidant compounds significantly increased [61].

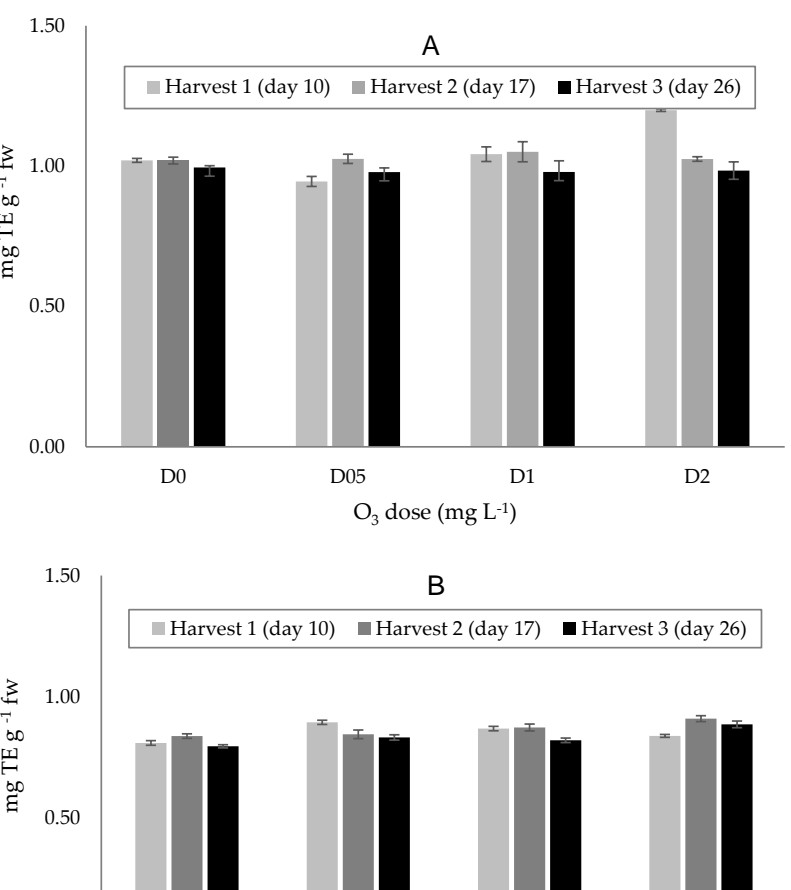

**Figure 4.** Antioxidant capacity via FRAP (**A**) and DPPH (**B**) methods expressed as mg Trolox eq (TE)$\cdot$g$^{-1}$ fw, of hydroponic baby red chard cv. SCR 107 exposed to four O$_3$ doses and three consecutive harvests. Values represent the mean ($n$ = 3) $\pm$ standard error.

### 3.4. Oxidative Stress Enzymes

3.4.1. Superoxide Dismutase

According to the statistical analysis, differences were found in the dose factor (Table 9), but the harvest factor showed no differences. It was found that SOD activity increased significantly with the application of O$_3$ compared to the control (Table 9) with values between 6.32 (D05) and 2.42 AU mg$\cdot$protein$^{-1}$ (D0). This increase would show that the O$_3$ application increased SOD activity due to oxidative stress. Plants have well-developed defense systems against stress and SOD would be the first defense against oxidative stress caused by O$_3$, as it is responsible for the dismutation of O$_2{}^-$ into H$_2$O$_2$ and O$_2$. Subsequently, enzymes such as CAT and APX could destroy this free radical [64]. SOD was found to be increased in Arabidopsis (*Arabidopsis thaliana* L.), rice (*Oryza sativa* L.), and Aleppo pine (*Pinus halepensis* Mill.) leaves exposed to O$_3$ [65,66]. In a study with *Pinus halepensis*, with applications of 200 g$\cdot$L$^{-1}$ of O$_3$ for 6 h per day for 8 days, higher SOD activity was observed in younger leaves, which would indicate that the defense system probably depends in turn on the age of the plant [65]. In another study, a 20% increase in SOD activity was obtained in spinach leaves treated with a total O$_3$ of 15,048 nL$\cdot$L$^{-1}\cdot$h$^{-1}$ over a 2-month period [67].

**Table 9.** SOD enzyme activity (AU mg·protein$^{-1}$) of hydroponic baby red chard cv. SCR 107 exposed to four $O_3$ doses.

| $O_3$ Dose (mg·L$^{-1}$) | SOD (AU mg·protein$^{-1}$) |
| --- | --- |
| D0 | 2.42 b [1] |
| D05 | 6.32 a |
| D1 | 5.11 a |
| D2 | 6.20 a |

[1] Different letters indicate significant differences according to Tukey's test (*p*-value ≤ 0.0001). Values represent the means (*n* = 9) of the three harvest times.

### 3.4.2. Catalase

Significant differences were found in the harvest factor. In each harvest, the highest CAT activity was observed with the highest doses applied (D1 and D2). In harvest 1, no significant differences among doses were observed (Figure 5A); in harvest 2, treatments D1 and D2 differed from the control; while in harvest 3, treatments D1 and D2 also differed significantly from D0 and D05. The above reaffirms that in the face of stress caused by $O_3$ application, chard plants activated their defense systems, as evidenced by the increase in CAT activity compared to the control at each harvest and, in turn, at higher doses of $O_3$. On the other hand, it has been shown that the response to oxidative stress caused by $O_3$ can vary within the same species. In a study on two varieties of beans (*Phaseolus vulgaris* L.), Irai and Fepagro 26 were exposed to 0.212 mg·L$^{-1}$·h$^{-1}$ of $O_3$ for one week, and CAT activity was found to vary according to the variety. The Irai variety significantly increased the activity of this enzyme, whereas Fepagro 26 remained significantly unchanged, indicating that the Irai variety has a better cellular capacity to reduce $H_2O_2$ [68].

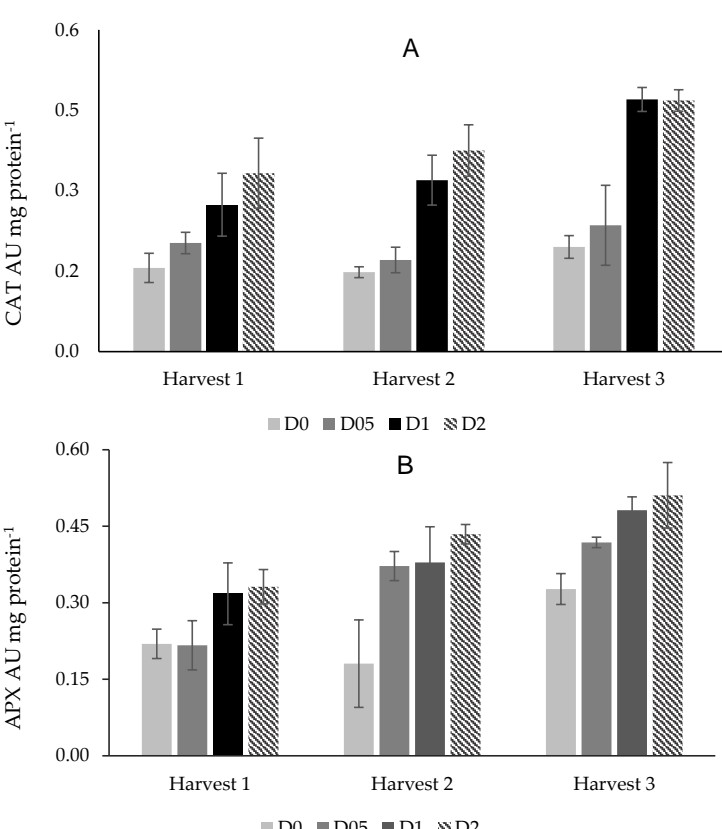

**Figure 5.** CAT (**A**) and APX (**B**) enzyme activities (AU mg·protein$^{-1}$) of hydroponic baby red chard cv. SCR 107 exposed to four $O_3$ doses and three consecutive harvests. Values represent the mean (*n* = 3) ± standard error.

### 3.4.3. Ascorbate Peroxidase

Following the same trend described for SOD and CAT, the APX activity showed an increase under the highest doses (D1 and D2); however, only at harvest 3 were significant differences observed between D2 and D0 (Figure 5B). Higher APX activity could contribute to the protection against ROS generated under different types of stress situations [69]. In this sense, in wheat research, higher APX activity was observed in those cultivars tolerant to heat stress [70]. The induction of APX as a response to stress has been described in various species. Lemoine et al. [30] observed significantly higher APX activity in broccoli treated with a postharvest UV-C stress treatment similar to the increase caused by $O_3$ applications. Similarly, Wang et al. [71] observed a 30% increase in APX after applications of 0.012 $mg \cdot L^{-1}$ of $O_3$ on rice due to oxidative stress caused by this gas. Also, Sarkar et al. [69] observed a 43.1% increase in APX in rice exposed to 0.004 $mg \cdot L^{-1}$ of $O_3$ daily during cultivation. The same was observed in a study on spinach leaves exposed to 15.048 $nl \cdot L^{-1} \cdot h^{-1}$ of $O_3$ for 2 months with a 36% increase in APX activity [67].

There are numerous studies describing an increase in the activity of antioxidant enzymes due to exposure of plants to $O_3$ [72–74]. In another study, the activity of the guaiacol peroxidase enzyme in plants treated with 0.048 $mg \cdot L^{-1}$ of $O_3$ for one month was 4 times higher than in control plants [75]. According to Liu et al. [76] doses of 40 ppb of ozone increased APX and CAT activities in wheat plants compared to 120 ppb and control. Although all the studies reviewed have shown that $O_3$ can induce an increase in one or more antioxidant enzymes such as SOD, CAT, and/or APX, no single effect has been determined in the same species. In fact, differential sensitivity to $O_3$ exhibited even by different varieties within the same plant species has been observed, which apparently depends on the efficiency of ROS scavenging [10,14,61].

To understand this phenomenon, it is worth mentioning that in normal plant cell metabolism, ROS formation takes place as a consequence of the damage caused by oxidative stress, a process in which SOD, APX, and CAT are involved. However, the degree of plant sensitivity to $O_3$ would also depend on both its natural antioxidant content and the intensity of the stress [62].

### 4. Conclusions

According to the results, for hydroponic baby red chard cv. SCR 107 a safe upper operating limit was 0.5 $mg \cdot L^{-1}$ of $O_3$, applied for 3 min every 2 days directly to the nutrient solution, inducing an increase in fresh matter and leaf area. Doses above this limit do not result in a significant yield increase and can be phytotoxic, showing visible damage such as chlorosis and leaf necrosis. The application of $O_3$ to the nutrient solution at 0.5, 1, and 2 $mg \cdot L^{-1}$ caused oxidative stress reflected in increased superoxide dismutase, catalase, and ascorbate peroxidase activity. The application of $O_3$ did not generate an impact on the macronutrient contents of the nutrient solution; however, micronutrients such as Fe and Mn contents decreased significantly, so the implementation of a replacement should be considered.

Therefore, the application of ozone at low doses of 0.5 $mg \, L^{-1}$, as a sanitization method for closed recirculating hydroponic systems would be a safe, viable, simple, and economical alternative to implement, particularly for small farmers. Foliar applications on leaf crops could be used to correct Fe or Mn deficiencies.

**Author Contributions:** Conceptualization, A.M.V. and A.C.S.G.; methodology, A.M.V.; formal analysis, C.H.-A. and A.C.S.G.; investigation, A.M.V.; resources, V.H.E.C.; data curation, C.H.-A. and A.M.V.; writing—original draft preparation, A.M.V., A.C.S.G. and V.H.E.C.; writing—review and editing, C.H.-A. and V.H.E.C.; project administration, V.H.E.C.; funding acquisition, V.H.E.C. All authors have read and agreed to the published version of the manuscript.

**Funding:** This research was funded by PROYECTO FONDECYT (ANID, Chile) number 1120274. The doctoral scholarship granted to Ms. A. Machuca by ANID number 21120299 is also appreciated.



**Data Availability Statement:** No new data were created or analyzed in this study. Data sharing is not applicable to this article.

**Conflicts of Interest:** The authors declare no conflict of interest.

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
