# Peer review of "Effect of the Ozone Application in the Nutrient Solution and the Yield and Oxidative Stress of Hydroponic Baby Red Chard"

_horticulturae, doi:10.3390/horticulturae9111234_

Round 1

Reviewer 1 Report

The author conducted a detailed investigation on the effects of applying different amounts of ozone on the yield, nutrient composition, antioxidant compounds, and oxidative stress enzymes of hydroponic red chard, which has certain guiding significance for hydroponic production. But before it can be accepted for publication, there are several issues that must be overcome.

1. The introduction section must be rewritten. There are too many lengthy and difficult to understand sentences. I have highlighted some in the manuscript.

2. Why did an independent experiment be conducted instead of measuring changes in nutrient solution components in the cultivation experiment?(see 2.3. Determinations of the Nutritive Solution)

3. It is difficult for me to understand that there was no significant difference in the biomass of hydroponic baby chard harvested at different times. In theory, as the cultivation time increases, the biomass should rapidly increase. The author did not provide necessary explanations either (see 3.1. Evaluation of Plant Growth).

4. The data in Table 6 is not the average of the three harvests in Table 5, and the author should carefully examine it.

5. The results section is too long and extensive, and the author should carefully reorganize the research results.

6. It is necessary for the author to unify the expression style in the text, for example, Figure 4 and Figure 5, baby chard and red chard.

1. The introduction section must be rewritten. There are too many lengthy and difficult to understand sentences. I have highlighted some in the manuscript.

Author Response

Dear Editor

We greatly appreciate the opportunity to publish our work in the Horticulturae journal.

We have made a thorough review of the paper and its results and considered the comments of both reviewers. All rewritten sentences were marked in red for ease of review by the reviewers

Below we responded to each of the reviewers' comments.

Reviewer 1:

Comment 1. The introduction section must be rewritten. There are too many lengthy and difficult to understand sentences. I have highlighted some in the manuscript.

Answer 1. The abstract and introduction were revised and rewritten to shorten sentences and clarify ideas. Some of the changes were made in lines 12, 13, 16, 19, 20, 39, 43, 48, 49, 56, 60, 61, and 76, among others.

Comment 2. Why did an independent experiment be conducted instead of measuring changes in nutrient solution components in the cultivation experiment? (see 2.3. Determinations of the Nutritive Solution)

Answer 2. A paragraph was included in lines 159 and 160 where it is indicated that the measurement was performed independently to avoid interaction between plant tissues and ozone gas concentration.

Comment 3. It is difficult for me to understand that there was no significant difference in the biomass of hydroponic baby chard harvested at different times. In theory, as the cultivation time increases, the biomass should rapidly increase. The author did not provide necessary explanations either (see 3.1. Evaluation of Plant Growth).

Answer 3. In lines 255 to 260, a possible justification for the results related to the harvest index and the time of the year in which the crops were grown was added.

.

Comment 4. The data in Table 6 is not the average of the three harvests in Table 5, and the author should carefully examine it.

Answer 4. In Table 5 the values shown correspond to the average of three crop units. In the case of Table 6, the number of data considered for the mean was added (line 298).

Comment 5. The results section is too long and extensive, and the author should carefully reorganize the research results.

Answer 5. The wording of the results was revised and reorganized.

Comment 6. It is necessary for the author to unify the expression style in the text, for example, Figure 4 and Figure 5, baby chard and red chard.

Answer 6. All the texts of the figures and tables were revised and uniformed by placing the baby red chard. See lines 261, 267, 276, 281, 290, and 296 among others.

Reviewer 2 Report

This paper provides a comprehensive investigation into the effects of ozone (O3) application on the growth, physiological responses, and oxidative stress markers of red chard plants, presenting valuable insights into the potential benefits and challenges of utilizing ozone in hydroponic cultivation systems. The study delves into the intricate relationship between ozone dosages and their impact on various parameters, offering a systematic analysis of plant growth, chlorophyll content, antioxidant capacity, and oxidative stress enzymes over multiple harvests.

However, there are several comments and questions to address:

Comments:

           How does the study address existing knowledge gaps?

           Line 249: The heading should be written as "Results and Discussion" instead of solely "Results."

           Line 498: Write “Phaseolus vulgaris” in italics.

           The discussion lacks a comprehensive interpretation of the results. For instance, the interpretation of chlorophyll changes and their relationship with yield and oxidative stress is missing.

           In conclusion, discuss the practical implications of the results for hydroponic crop cultivation.

           The paper requires thorough proofreading. Some sentences are challenging to understand. For example, lines 342-346.

Questions:

1-         Could you elaborate on the reason for selecting a floating hydroponic system and the chosen crop (baby chard) for this study? How representative is this system for broader hydroponic practices?

2-         How was the ozone application process controlled to ensure consistent and accurate dosing throughout the study? Were factors like air bubbles or variations in ozone concentration considered?

3-         May I inquire about the explanation for the exclusion of carotenoid content measurement in the conducted study, considering their importance in oxidative stress mitigation and their involvement in various physiological processes?

4-         Were the ozone concentrations (0.5, 1.0, and 2.0 mg·L-1) selected based on previous research or safety considerations?

5-         Given the observed decrease in Fe and Mn concentrations in the nutrient solution after ozone application, could you speculate on potential methods to mitigate these deficiencies while still benefiting from ozone's sanitizing effects?

6-         The paper mentions the increase in oxidative stress enzymes (SOD, CAT, and APX) in response to ozone application. Could you provide more insight into the implications of this increase on plant health and growth, and how it relates to the observed yield changes?

7-         In the discussion section, could you elaborate on the potential practical applications of the findings in terms of optimizing ozone application for hydroponic chard production? How can farmers use this information to improve crop yields and food safety?

Minor editing of English language required

Author Response

Dear Editor

We greatly appreciate the opportunity to publish our work in the Horticulturae journal.

We have made a thorough review of the paper and its results and considered the comments of both reviewers. All rewritten sentences were marked in red for ease of review by the reviewers

Below we responded to each of the reviewers' comments.

Reviewer 2

Comment 1.   How does the study address existing knowledge gaps?

Answer 1. An important aspect of this new study is its focus on the effects on antioxidant compounds and the response of the plant antioxidant enzyme system to stress caused by direct application of ozone as a sanitizer in closed hydroponic systems.  Previous work cites the effect on plants of atmospheric ozone as a consequence of environmental pollution. See lines 475 to 476.

 Comments 2  Line 249: The heading should be written as "Results and Discussion" instead of solely "Results." and      Line 498: Write “Phaseolus vulgaris” in italics.

Answer 2. All these changes were made in the text.

Comment 3.   The discussion lacks a comprehensive interpretation of the results. For instance, the interpretation of chlorophyll changes and their relationship with yield and oxidative stress is missing.

Answer 3. With respect to this comment, it seems to us that an adequate discussion of the results has been made. In the specific case of the relationship between chlorophyll results and performance in the text this is discussed between lines 342 and 374 but especially 347 and 350.

Comment 4. In conclusion, discuss the practical implications of the results for hydroponic crop cultivation.

Answer 4. The following paragraph was added in the conclusions ‘Therefore, the application of ozone at low doses of 0.5 mg L-1, as a sanitization method for closed recirculating hydroponic systems would be a safe, viable, simple, and economical alternative to implement, particularly for small farmers. Foliar applications on leaf crops could be used to correct Fe or Mn deficiencies.’

Comment 5.    The paper requires thorough proofreading. Some sentences are challenging to understand. For example, lines 342-346.

Answer 5. The article was revised and its wording improved to make it clearer to understand.

Questions:

1-         Could you elaborate on the reason for selecting a floating hydroponic system and the chosen crop (baby chard) for this study? How representative is this system for broader hydroponic practices?

Baby red chard is a relevant product for the preparation of fresh-cut salads due to its appearance and color. Due to its nature and production systems, it can be easily contaminated by pathogenic bacteria for the consumer. The results obtained in this study can be considered for applications in other similar crops such as lettuce, arugula and spinach.

2-         How was the ozone application process controlled to ensure consistent and accurate dosing throughout the study? Were factors like air bubbles or variations in ozone concentration considered?

To regulate the application conditions of ozone, small 1L containers were used. Measurements were taken at noon to ensure consistent temperatures during each reading. The semi-industrial ozone generator was used to generate up to 30 g per hour with a sufficient flow to saturate the solution within 3 minutes. 

3-         May I inquire about the explanation for the exclusion of carotenoid content measurement in the conducted study, considering their importance in oxidative stress mitigation and their involvement in various physiological processes?

The measurement of the carotene content was not taken into consideration since the focus was on the measurement of total phenolic compounds and antioxidant capacity, in order to determine the impact of ozone on antioxidant compounds.

4-         Were the ozone concentrations (0.5, 1.0, and 2.0 mg·L-1) selected based on previous research or safety considerations?

The treatments were applied in 1L containers located in a 3 x 33 m chapel-type greenhouse to avoid risks to people. Considering the plants, a previous test was carried out to establish the ozone doses used.

5-         Given the observed decrease in Fe and Mn concentrations in the nutrient solution after ozone application, could you speculate on potential methods to mitigate these deficiencies while still benefiting from ozone's sanitizing effects?

As mentioned in the paragraph that was added to the conclusion, a possibility to mitigate the decrease of Fe and Mn would be the foliar applications of these microelements.

6-         The paper mentions the increase in oxidative stress enzymes (SOD, CAT, and APX) in response to ozone application. Could you provide more insight into the implications of this increase on  plant health and growth, and how it relates to the observed yield changes?

In relation to this question, for each enzyme analyzed, a discussion was made with the results of other studies. At the end of this section, it was mentioned how the response to ozone depends on the sensitivity and the natural antioxidant content of the plant, the doses, among other factors.

7-         In the discussion section, could you elaborate on the potential practical applications of the findings in terms of optimizing ozone application for hydroponic chard production? How can farmers use this information to improve crop yields and food safety?

As mentioned in the concluding paragraph, the application of ozone in low doses through simple injection systems can be a cost-effective and safe alternative for growers. To minimize risks, these applications could be done during nighttime when there are no people, or in periods of increased greenhouse ventilation. The ozone application could be performed in the nutrient solution's reservoir pond located outside the greenhouse.

Round 2

Reviewer 1 Report

The revised manuscript is generally satisfactory, but there are still some issues. 1. The Dry weight(%) in the header of Table 5 should be “Leaf area(cm2)”.  2. Since the data in Table 6 is the means (n=9) of the three consecutive harvests, it should be at least close to the average of the three corresponding data in Table 5. However, there is clearly an obvious error in the data in Table 6. According to the data in Table 6, the leaf area of D05 is the lowest, which is clearly inconsistent with the data in Table 5.

There are still some difficult to understand sentences, such as , However, according to Risoli and Lauria [14] despite the advantages of O3 compared to other sanitization methods it is recognized as a phytotoxic gas.【line63-64】

Author Response

Dear reviewer.
We appreciate the comments you have made on our work. The error in table 5 was corrected and marked in green.
In addition, a paragraph in green was added on lines 274 to 275 to better explain the contents of table 3. 
In relation to the paragraph where Risoli and Lauria are mentioned, the word 'it' was deleted to try to clarify the idea.

Best Regards,

Reviewer 2 Report

I appreciate the author's efforts in addressing some of the comments in my review. Their revision have certainly changed the overall quality of the manuscript. However, I would like to kindly request further clarification on some of the questions I had posed, which, as of the current revisions, remained unanswered.

Questions:

1-      Could you elaborate on the reason for selecting a floating hydroponic system and the chosen crop (baby chard) for this study? How representative is this system for broader hydroponic practices?

2-      How was the ozone application process controlled to ensure consistent and accurate dosing throughout the study? Were factors like air bubbles or variations in ozone concentration considered?

3-      May I inquire about the explanation for the exclusion of carotenoid content measurement in the conducted study, considering their importance in oxidative stress mitigation and their involvement in various physiological processes?

4-      Were the ozone concentrations (0.5, 1.0, and 2.0 mg·L-1) selected based on previous research or safety considerations?

Author Response

Dear Mr/Ms reviewer.
We have tried our best to respond to your comments. We agree with your observations but there are parameters that we can no longer evaluate at this stage.
I have completed the answers I gave you previously. I hope you like them and that we can publish our work in Horticulturae.

Comments:

1. Could you elaborate on the reason for selecting a floating hydroponic system and the chosen crop (baby chard) for this study? How representative is this system for broader hydroponic practices?

Answer 1.  As part of the reasons we added before, there are many studies mentioned among closed-loop hydroponic techniques, the floating system is widely used for the cultivation of leafy vegetables with short cycles and grown at high plant density, such as many baby-leaf vegetables and herbs including sweet basil and lettuce.

  1. Kotsiras, A. Vlachodimitropoulou, A. Gerakaris, N. Bakas, A.I. Darras. 2016. Innovative harvest practices of Butterhead, Lollo rosso and Batavia green lettuce (Lactuca sativa L.) types grown in floating hydroponic system to maintain the quality and improve storability. Scientia Horticulturae 201 (30): 1-9
  2. Landi, A. Pardossi, D. Remorini, L. Guidi. 2013. Antioxidant and photosynthetic response of a purple-leaved and a green-leaved cultivar of sweet basil (Ocimum basilicum) to boron excess. Environ. Exp. Bot. 85: 64-75
  3. Kiferle, R. Ascrizzi, M. Martinelli, S. Gonzali, L. Mariotti, L. Pistelli, G. Flamini, P. Perata. 2019. Effect of Iodine treatments on Ocimum basilicum L.: Biofortification, phenolics production and essential oil composition. PLoS One, 14: 1-23

2. How was the ozone application process controlled to ensure consistent and accurate dosing throughout the study? Were factors like air bubbles or variations in ozone concentration considered?

Answer 2. As we tried to explain before the O3 content was determined with a portable O3 analyzer (Chemetrics, model I-2019, USA) similar to the one it showed in the link https://www.oxidationtech.com/i-2019.html.

Small 1L containers were used to regulate the application conditions of ozone. Measurements were taken at noon to ensure consistent temperatures during each reading. The semi-industrial ozone generator was used to generate up to 30 g per hour with a sufficient flow (6L / min) to saturate the solution within 3 minutes.

3. May I inquire about the explanation for the exclusion of carotenoid content measurement in the conducted study, considering their importance in oxidative stress mitigation and their involvement in various physiological processes?

Answer 3. Unfortunately, the measurement of the carotene content was not taken into consideration because as we mentioned before our focus was on the measurement of total phenolic compounds and antioxidant capacity and the oxidative enzymes to determine the impact of ozone on the stress response of the plants.

4-      Were the ozone concentrations (0.5, 1.0, and 2.0 mg·L-1) selected based on previous research or safety considerations?

Answer 4. The previous trial was made to choose adequate ozone doses is described between lines 106 to 110. The treatments were applied in 1L containers located in a 3 x 33 m chapel-type greenhouse to avoid risks to people. Considering the plants, a previous test was carried out to establish the ozone doses used.

Round 3

Reviewer 1 Report

I have previously mentioned the issue of data errors in Table 6, but the author clearly overlooked this issue and I don't know why. If the author could be a little more careful, they could find that the data in Table 4 and Table 6 are exactly the same, which is clearly an unacceptable error.

Please carefully check the grammar errors in the following sentence.

However, according to Risoli and Lauria [14] despite the advantages of O3 compared to other sanitization methods is recognized as a phytotoxic gas  

Author Response

Dear Reviewer

I really appreciate your comments. All were attended and remarked in blue color

Sincerely

Victor

Reviewer 2 Report

  I think the authors have adequately addressed the comments, and I appreciate their efforts. The overall quality of the manuscript has definitely improved as a result of their revisions, therefore, I have no further comments and the paper is acceptable in present form.

Author Response

Dear Reaviewer

I really appreciate your comments and all of them were attended.

Sincerey,

Victor
